# Effect of Ni(P) Layer Thickness on Interface Reaction and Reliability of Ultrathin ENEPIG Surface Finish

**DOI:** 10.3390/ma14247874

**Published:** 2021-12-19

**Authors:** Panwang Chi, Yesu Li, Hongfa Pan, Yibo Wang, Nancheng Chen, Ming Li, Liming Gao

**Affiliations:** School of Material Science and Engineering, Shanghai Jiao Tong University, Shanghai 200240, China; CHIPANWANG@sjtu.edu.cn (P.C.); liyesu@sjtu.edu.cn (Y.L.); phf@alumni.sjtu.edu.cn (H.P.); 19960224hl@sjtu.edu.cn (Y.W.); hi2018.charles@gmail.com (N.C.); mingli90@sjtu.edu.cn (M.L.)

**Keywords:** surface finish, ultrathin ENEPIG, growth of IMC, reliability

## Abstract

Electroless Ni(P)/electroless Pd/immersion Au (ENEPIG) is a common surface finish in electronic packaging, while the Ni(P) layer increases the impedance of solder joints and leads to signal quality degradation in high-frequency circuits. Reducing the thickness of the Ni(P) layer can balance the high impedance and weldability. In this paper, the interfacial reaction process between ultrathin ENEPIG substrates with different Ni layer thicknesses (0.112 and 0.185 μm) and Sn–3.0Ag–0.5Cu (SAC305) solder during reflow and aging was studied. The bonding ability and reliability of solder joints with different surface finishes were evaluated based on solder ball shear test, drop test and temperature cycle test (TCT), and the failure mechanism was analyzed from the perspective of intermetallic compound (IMC) interface growth. The results showed that the Ni–Sn–P layer generated by ultrathin ENEPIG can inhibit the growth of brittle IMC so that the solder joints maintain high shear strength. Ultrathin ENEPIG with a Ni layer thickness of 0.185 μm had no failure cracks under thermal cycling and drop impact, which can meet actual reliability standards. Therefore, ultrathin ENEPIG has broad prospects and important significance in the field of high-frequency chip substrate design and manufacturing.

## 1. Introduction

With the development of the electronic information industry, electronic packaging technology is focusing on the direction of high function, high density and high integration [1,2]. Solder joints play an important role not only in mechanical connection but also in electrical connection in electronic packaging. Therefore, solder joint reliability is an important research content of modern electronic packaging technology [3,4,5]. In recent years, the surface finish of printed circuit boards (PCBs) has attracted more and more attention because of its protective effect on exposed copper circuits and for providing a good soldering surface, which affects the reliability of electronic packaging solder joints [6,7,8]. Compared with common surface finishes such as Organic Solderability Preservative (OSP), Immersion Tin (ImSn), Nickel–Gold (NiAu) and Electroless Nickel/Immersion Gold (ENIG), ENEPIG has the advantages of good thermal stability, high weldability, low cost and no black pad phenomenon, like ENIG [9,10,11,12,13], and can be applied to a variety of packaging forms. The Ni layer in ENEPIG is a widely used barrier material to prevent the rapid generation of brittle IMC from copper in the pad and tin in the solder [14]. The gold layer improves the wettability and prevents the oxidation of the pad. The Pd layer is an amorphous structure that prevents the rapid diffusion of nickel atoms, reduces the surface activation energy of the pad and improves the solderability of surface finish [15]. Studies have shown that during the welding process between SAC305 solder and ENEPIG, the Au layer and Pd layer will rapidly melt into the solder, and the solder will come into contact with the Ni(P) layer and form intermetallic compounds. As the soldering continues, Ni atoms diffuse into the solder and form (Pd, Ni) Sn_4_, (Cu, Ni)_6_Sn_5_ with Pd and Sn atoms. Due to the consumption of Cu atoms, they are transformed into (Cu, Ni)_3_Sn_4_ [16,17]. The existence of phosphorus causes the phenomenon of soldering reaction-assisted crystallization, which means that the amorphous Ni(P) alloy layer is gradually transformed into crystalline Ni_3_P [18].

In addition to paying attention to the interface reaction process of different surface finishes, many researchers have studied the reliability performance of ENEPIG and other surface finishes. Ruyu Tian et al. [19] compared the reliability performance of Sn–Ag–Cu/ENIG and Sn–Ag–Cu/ENEPIG solder joints under extreme temperature impact. The results showed that Sn–Ag–Cu/ENEPIG solder joints constantly have high shear strength and thermal impact resistance. Sun et al. [20] welded five kinds of surface finish pads with SAC305 solder, tested their drop characteristics and carried out board level temperature cycle experiments. Finally, it was found that an ENEPIG substrate showed excellent performance and good long-term reliability in the two reliability experiments.

However, due to the ENEPIG process, the chemically deposited Ni(P) alloy layer significantly increases the resistance of the solder joints. The resistivity of the amorphous Ni(P) layer is 70~120 μΩcm, which is an order of magnitude higher than that of other solder joints materials such as Cu (1.7 μΩcm), Sn (11.5 μΩcm) and crystalline Ni (6.8 μΩcm) [21]. High-impedance solder joints will reduce electrical performance and lead to signal quality degradation, which affects the signal integrity in integrated circuits. This is particularly obvious in high-frequency components.

A feasible method is to adjust the Ni layer thickness of ENEPIG surface finish to alleviate the above problems [22]. However, current research on the surface finish of ENEPIG is still mainly focused on the traditional ENEPIG of about 4~5 μm. Some researchers have begun to pay attention to the interface reaction process of ultrathin ENEPIG and the bonding force of solder joints. Hua Miao et al. [23] found that the ultra-thin ENEPIG with a thickness of 0.1~0.3 μm Ni(P) plating was consumed after reflowing with lead-free solder, and a stable IMC could be formed. Cheng-Ying Ho et al. [24] found that when the Ni(P) thickness was greater than the critical value (about 0.18 μ m), a P-rich IMC layer was formed; in addition, although the Ni(P) layer is exhausted in the initial stage of the reflow process, the formation of the Ni_3_P and Ni_2_Sn_1+x_P_1−x_ phases of the P-rich layer can still play the role of a diffusion barrier to prevent the rapid diffusion of Cu. Jong-Hoon Back et al. [25] compared the IMC growth process of thin ENEPIG and normal ENEPIG and their solder joints’ adhesion under different speed shear tests. The results showed that there was no significant difference between the two surface finishes in low-speed shear tests, but because the IMC layer of thin ENEPIG was thicker than that of normal ENEPIG, the adhesion of thin ENEPIG was poor in high-speed shear. Research on board-level reliability has recently emerged, but the micro-mechanism is still a challenge.

In this study, different surface finishes such as ultrathin ENEPIG, OSP, electroplated NiAu and conventional ENEPIG were compared. The interface reaction process between ultrathin ENEPIG surface finish and lead-free solder was studied through aging experiments, and the solder joints’ adhesion and the board level reliability of different surface finishes were compared. The results show that the ultrathin ENEPIG can be taken into account for its barrier effect and board level reliability. This research provides an experimental and theoretical basis for the practical application of ultrathin ENEPIG, which is of great significance to the design of high-frequency chips.

## 2. Materials and Methods

This paper first studied the interface reaction process between ENEPIG and SAC305 solder with different Ni(P) layer thicknesses and compared this with OSP substrate. The purpose of setting the electroplated Ni/Au surface finish was to compare the board-level reliability at a later point. For different samples of ENEPIG and electroplated NiAu substrates, the actual value of coating thickness is measured by using an X-ray fluorescence spectrometer (AHIMADZU, Kyoto, Japan). The actual value of coating thickness is shown in Table 1, and the surface morphology of coating is shown in Figure 1. ENEPIG 1 and ENEPIG 2 are ultrathin Ni (P) layers, and ENEPIG 3 is a normal coating.

The aging test after reflowing the solder balls can accelerate the progress of the IMC interface reaction. The structure of the solder joints after reflowing is shown in Figure 2. The aging temperature selected in this paper based on the JESD22-A103D standard was kept constant at 150 °C, and there was nitrogen atmosphere in the furnace to prevent oxidation of the samples. The morphology and evolution process of IMC with different surface finishes were studied by observing the cross-sections of the samples by SEM (TESCAN, Brno, Czech Republic). Based on an energy dispersive spectrometer (EDS, TESCAN, Brno, Czech Republic), the characteristic X-ray intensity and wavelength excited by different elements in the IMC layers were received and matched with the corresponding energy spectrum to determine the elements and content of the IMC layer.

As the morphology of the interface IMC in this experiment was uneven, the thickness of IMC layer is defined as follows [26,27]:T = S/L(1)
where T is thickness of IMC layers, S is area of IMC layer and L is length of IMC layer. In this paper, the area of the IMC layer was measured by using ImgaeJ 1.8.0 (National Institutes of Health, Bethesda, MD, USA) to recognize the contrast of the IMC layer. The length of the IMC layer was measured by using image rulers and a scale bar in SEM (TESCAN, Brno, Czech Republic) images. The average thickness (T_ave_) is the average of the thicknesses of 10 solder joints in test samples.

The solder ball shear experiment, based on the JESD22-B117A standard, evaluated the bonding ability of different Ni layer thicknesses’ surface finishes and solder joints in different aging stages in which the shear speed was 0.01 m/s and the shear height was 30 μm. The board level drop test standard used in this experiment was JESD22-B111; the maximum G value is 1500 g, the drop time is 0.5 ms and the failure judgment criterion is that the resistance exceeds 1000 ohm for four consecutive times. The high- and low-temperature cycle test standard used in the TCT was Condition G in JESD22-A104E; the temperature ranges from −40 °C to 125 °C, the holding time and heating time are 15 min each and the failure judgment criterion is that the loop resistance exceeds 1000 ohm.

## 3. Results and Discussion

### 3.1. The Growth of Interface IMC during Aging

This section mainly discusses the morphology and evolution of IMC after reflow and aging with different surface finishes. As mentioned above, much literature [21,22,23,28] has studied the IMC composition of standard ENEPIG and OSP. In this section, the composition changes in IMC of different surface finish substrates are calculated by EDS analysis (TESCAN, Brno Czech Republic) and summarized in the cross-section of the IMC evolution process. This paper focuses on the changes in IMC composition during the interface reaction of the ultrathin Ni(P) layer ENEPIG, which has previously been studied less.

For OSP substrates, because there is no Ni layer barrier during the reflow process, copper atoms react with Sn in the solder at the interface to form IMC. A part of the copper involved in the reaction derives from the solder, and the other part derives from the diffusion of copper in the pad. The IMC interface reaction process of the OSP substrates is shown in Figure 3.

Firstly, the uneven Cu_6_Sn_5_ intermetallic compound was formed at the interface. With the extension of aging time, the thickness of Cu_6_Sn_5_ increased, the morphology changed from scallop to plane layer and a new IMC Cu_3_Sn layer was gradually formed below Cu_6_Sn_5_. The interface of the Cu_6_Sn_5_/Cu_3_Sn/Cu sandwich structure was formed, which is in line with the results of previous experiments. As Cu_3_Sn possesses great brittleness compared with solder and other types of IMC, the influence of Cu_3_Sn on the properties of solder joints will be discussed in subsequent chapters.

The reaction process of the interface IMC layer during aging of ultrathin ENEPIG is shown in Figure 4 and Figure 5. As the gold layer and palladium layer on the ENEPIG substrate will rapidly diffuse into the solder during reflow to form dispersed IMC, the main factors affecting the growth of IMC at the interface are the diffusion of Sn-Cu and the barrier effect of the Ni(P) layer.

The IMC on the ultrathin ENEPIG substrate presents fibrous protrusion in comparison with the scallop shaped IMC on OSP substrate. In addition, the overall morphology of IMC tends to be flat during the progress of interface reaction, with the thickness gradually increasing.

In the experiment, the thickness of the Ni(P) layer of ENEPIG 1 and ENEPIG 2 was very thin, being only about 0.1 μm. During the reflow process, the Ni(P) layer reacts rapidly with the liquid solder, making it difficult to observe the existence of the Ni layer after reflow. However, without the diffusion of the P atoms in the Ni(P) alloy layer, the Sn diffused from the solder to the Ni(P) layer will react with the remaining Ni(P) layer to form Ni–Sn–P, as shown in Figure 4c and Figure 5c. After 120 h of aging, a thin layer of discontinuous flake IMC was formed on the interface between ENEPIG 1 and ENEPIG 2, and the IMC was (Cu, Ni)_3_Sn through EDS composition analysis, as shown in Figure 6 and Figure 7. Furthermore, (Cu, Ni)_3_Sn gradually tended to be continuous and thicker as the aging experiment progressed.

The energy spectrum analysis results of the interface IMC of the 120 h ENEPIG 2 are summarized in Table 2. The IMC of points A and B was approximately (Cu, Ni)_6_Sn_5_, and the IMC of point C was (Cu, Ni)_3_Sn. In Table 2, it can be found that a small amount of P and Pd was detected at points A and B, separately. The blocking Ni–Sn–P layer may further react with the solder and be consumed, leading to the detection of the small amount of P. Moreover, Pd rapidly diffused into Sn in the reflow process to form a dispersed distribution, which explains the reason for the existence of Pd. During the progress of the interface reaction, Pd can diffuse into the interface IMC.

When the aging time reached 256 h, after calculation, the IMC of point A and point B remained as (Cu, Ni)_6_Sn_5_, however, the content of Ni in the newly generated IMC of point C was only 0.1%. Due to the massive consumption of the Ni–P layer, the Ni that participated in the reaction stably exists in the previously generated IMC with little Ni diffusing to the interface for the reaction.

As the Ni(P) layer has an inhibitory effect on Cu diffusion, the thinner Ni(P) layer deposited on ENEPIG 1 and ENEPIG 2 was quickly consumed during reflow and aging, which weakened the inhibitory effect on Cu diffusion. As a result, there exists a brittle (Cu, Ni)_3_Sn IMC phase below (Cu, Ni)_6_Sn_5_ in ultrathin ENEPIG, which is similar to that observed for the OSP substrate. In contrast, ENEPIG 3 had a strong inhibitory effect on Cu diffusion due to the deposition of a thick Ni(P) layer, and there was still no discontinuous brittle (Cu, Ni)_3_Sn at 500 h, as shown in Figure 8.

The aging time was extended to 1000 h for further observation of the IMC growth of the three ENEPIGs and the OSP under long-term reaction conditions, and the results are shown in Figure 9.

After long-term aging, the surface morphology of IMC with different surface finishes tended to be flat, and the IMC at the ENEPIG interface gradually changed from thin rod to layered. From the perspective of IMC growth thickness, the total thickness of OSP and ultrathin ENEPIG IMC was similar, while the brittleness and (Cu, Ni)_3_Sn IMC thickness of the ultrathin ENEPIG interface were much smaller than those of OSP. Therefore, although the inhibitory effect on Cu diffusion was gradually weakened due to the depletion of the Ni(P) layer, the Ni–Sn–P layer formed by the reaction still inhibited Cu diffusion.

For ENEPIG with the standard Ni layer thickness, the IMC layer with the smallest total thickness was formed after 1000 h, in which the continuously layered, brittle (Cu, Ni)_3_Sn was still not found, and there was still no completely consumed Ni(P) layer at the interface. Therefore, ENEPIG with standard Ni layer thickness has the best inhibitory effect on Cu diffusion.

Sn solder balls were corroded by using an alcohol solution of 4% HNO_3_ + 1% HCl, and the upper surfaces of IMC at different surface finish interfaces after aging for 64h were observed. The SEM results are shown in Figure 10. The observation results of the upper surface of the interface IMC layer are consistent with that of the cross-section of the interface IMC during the previous aging process. Cu_6_Sn_5_ IMC that is formed at the OSP interface presents a massive bulge, whose growth is relatively uneven. For ultra-thin ENEPIG, fibrous (Cu, Ni) _6_Sn_5_ IMC with a random extension direction appears; for ENEPIG with a high Ni layer thickness, the fibrous result is not entirely obvious due to the slow growth of IMC.

The interface reaction processes of several surface finishes were sorted out, and IMC growth models of OSP, ultrathin ENEPIG and standard ENEPIG were proposed, as shown in Figure 11. The element diffusion rate at Sn solder/pad interface was the main factor affecting the IMC evolution of different surface finishes.

As there was no Ni barrier layer in OSP, the scallop shaped Cu_6_Sn_5_ IMC was formed by a rapid reaction between Sn and Ni after refluxing. With the extension of aging time, the morphology of Cu_6_Sn_5_ IMC changed from scallop to plane layer, and a new IMC Cu_3_Sn was formed below Cu_6_Sn_5_. Fiber-like (Cu, Ni)_6_Sn_5_ IMC was formed in ENEPIG due to the existence of a Ni barrier layer, and a discontinuous brittle (Cu, Ni)_3_Sn IMC layer gradually appeared with the aging reaction. For ultrathin ENEPIG, the thickness of the electroless Ni(P) layer was less than 1 μm. Ni was rapidly consumed during reflow and short-term aging, however, the thin Ni–Sn–P layer formed by interfacial reaction still possessed a partial blocking effect on Cu diffusion.

### 3.2. Kinetic Study of IMC Growth

The relationship between IMC layer thickness and aging time can generally be expressed by the following equation [29]:(2)$$T=\sqrt[{n}]{\mathrm{Dt}}$$
where T is the thickness of IMC, D is the diffusion coefficient, t is the aging time, and n is the power-law index.

For the surface treated samples with different Ni layer thicknesses, the IMC thickness of the interface after different aging stages over 0–1000 h was measured, and the relationship between the change in IMC thickness and time was analyzed. The results are shown in Figure 12.

During the aging process, the thickness growths of all surface finishes of IMC were parabolic, which conforms to the law that the interface reaction is controlled by substance diffusion. Due to the addition of the Ni layer for blocking, it can be seen that the total thickness of the IMC layer and the brittle IMC layer of the ultrathin ENEPIG and ENEPIG decreased, and the brittle IMC layer never appeared in ENEPIG, which has the best effect on suppressing the diffusion of Cu at the interface.

Through the IMC growth thickness and the square root of aging time curves, two distinct and obvious stages of IMC growth for OSP and ENEPIG can be observed, which both meet the linear relationship between thickness and square root of aging time. From the previous discussion, we know that these two stages are the independent growth stage of Cu_6_Sn_5_ and the competitive stage of Cu_6_Sn_5_ consumption and Cu_3_Sn growth, separately. However, in the initial aging stage, a small step with a very low IMC growth rate appears in ENEPIG, which is also due to the blocking effect of Ni layer.

### 3.3. Shear Test Results

After the shear test of solder joints with different surface finishes after reflow and aging, there were four different failure modes, as shown in Figure 13:

It is generally believed that the fracture position that appears at the IMC interface is a brittle fracture with a lower shear strength, while the fracture position that appears inside the solder ball is a ductile fracture with a higher shear strength. The solder joints with a ductile fracture have better interface bonding properties. The failure modes of different surface finishes under different aging times were counted, and the results are shown in Figure 14.

Although the Ni layer of ultra-thin ENEPIG reacted quickly, its brittle fracture mode appeared later. This is due to the secondary barrier effect of the Ni–Sn–P layer, which makes it better than OSP but worse than ENEPIG in terms of fracture mode. Compared with electroplated NiAu samples, the brittle fracture ratio of ultrathin ENEPIG after 1000 h of aging is slightly lower than that of electroplated Ni/Au, leading to a better fracture mode than that of electroplated NiAu. By comparing the two ultrathin ENEPIG fracture modes, no obvious difference was found in the experiment.

During aging, not only will the failure mode of the shear test gradually change, but the shear strength between the welding ball and surface finish will also change under different aging times. The summary is shown in Figure 15.

With the progress of the aging process, the shear strength of all surface finishes showed a trend of first increasing and then decreasing. In the initial stage of reflow, the interface reaction time is very short, and the IMC growth is insufficient. At this time, the bonding is weak, and the shear strength is low. As the aging process progresses, the IMC layer continues to grow and the bonding capacity between the interfaces gradually increases from the initial weak bonding to the highest at about 16 h. However, IMC itself is quite brittle compared with solder. The overall brittleness at the interface increases with the growth of IMC, resulting in a decrease in shear strength. This phenomenon is consistent with the results of fracture mode.

Comparing the shear strength curves of each surface finish under different aging times, it can be found that ENEPIG and Ni/Au with a Ni layer have higher shear strength than OSP. On the one hand, the Ni layer inhibits the growth of IMC and weakens the influence of brittle IMC. On the other hand, Cu-Ni-Sn ternary IMC has higher mechanical strength than Cu-Sn. In addition, the fibrous tubular IMC will increase the roughness of the interface and the surface area of the solder and IMC contact, forming more anchor points to hinder the relative movement between the solder and the IMC interface, thereby generating greater resistance during the shearing process. Finally, the interface bonding ability increases.

### 3.4. Drop Test Results

This paper used drop experiments to evaluate the impact resistance of different surface finishes. The experimental results are shown in Table 3. The total number of drop experiments was designed to be performed 30 times according to the actual application scenarios of the device.

The results showed that ENEPIG 2, ENEPIG 3 and OSP all passed the drop test, but ENEPIG 1 and Ni/Au failed in this test.

We can observe the interface of the drop test sample, and the result is shown in Figure 16. During the experiment, the IMC layer generated by the interface reaction of all the samples was relatively thin, being only 2~3 μm without the brittle IMC layer observed. The morphology and composition of the IMC were similar to the results of the aging experiment.

The failed samples were sectioned to observe the location of cracks at the solder joints, and the results are shown in Figure 17. The cracks in the solder balls of the failed samples in either electroplated Ni/Au or ENEPIG 1 appeared at the junction of the IMC and Cu pads, and the cracks continued to extend to the entire interface, indicating that interface IMC strength is the key to affecting the drop performance of different surface finishes.

In the drop test, when the inside of the solder ball is subjected to impact from the pad, the Sn crystal grains will form dislocations to absorb part of the impact energy, and the generation of dislocations will increase the tensile strength and yield strength of the solder itself. At the same time, the strain-strengthening mechanism [30] enables the solder to increase its own strength to be higher than the fracture strength of the IMC at a very high strain rate (1% s^−1^ to 10% s^−1^) in the drop test, resulting in failures that mostly occur in the brittle IMC layer, as shown in Figure 18.

In this experiment, the sample was connected with PCB by one-time reflow soldering, where the growth time of IMC was short. As OSP had no Ni barrier layer, the solder reacted rapidly with the Cu pad to form IMC of a certain strength, which had good drop resistance. For the NiAu samples, Cu diffusion was inhibited by the Ni barrier layer; therefore, IMC was not fully formed in a short time, and the strength was low, leading to a poor drop performance. Due to the addition of a Pd layer in ENEPIG, there will be a small amount of PD in the IMC layer; this improves the strength of the IMC layer, resulting in a better drop performance than NiAu.

For ultrathin ENEPIG, the Ni layer was exhausted after reflow due to its thinner thickness. Only the secondary barrier effect of the Ni–Sn–P layer inhibited the diffusion of Cu. The generated IMC strength was better than that of NiAu and ENEPIG, and the IMC strength containing Pd was higher than the conventional (Cu, Ni)_6_Sn_5_ IMC. Therefore, it can be considered that the ultrathin ENEPIG with a Ni(P) layer thickness of 0.183 μm had a certain degree of drop resistance, which can meet the reliability that electronic devices require. The reasons of ENEPIG 1 failing the drop test may be as follows: the uneven surface of the thin coating, the formation of holes or the uneven growth of IMC during the interface reaction process, which may easily cause stress concentration at the interface and lead to failure.

### 3.5. Temperature Cycle Test Results

During the TCT experiment, the temperature cycle range was from −40 °C to 125 °C. According to the actual device application requirements, the design cycle was repeated 125 times. The experimental results are shown in Table 4.

In this experiment, all the samples passed the requirements of 125 cycles, which shows that ultra-thin ENEPIG can meet the requirements of practical application. In order to further compare the difference in temperature cycling resistance of different surface finishes, the number of cycles of the samples was increased. It was found that all samples had an open circuit after 1602 cycles for ENEPIG 1 and after 1124 cycles for OSP. Due to the time cost required for the cycle, no more cycle experiments were carried out.

The IMC growth of the sample after TCT is observed in this section. As shown in Figure 19, due to the impact of thermal load for a long time period, the IMC growth was sufficient, and its thickness was significantly higher than that of the drop test.

Combining the results of the temperature cycling experiment with IMC growth, it can be observed that there is no Ni barrier layer in OSP. When IMC grows rapidly, the volume mismatch caused by the CTE difference will produce a large amount of stress to promote the generation and extension of microcracks, meaning that the performance of TCT is the worst. In ENEPIG, due to the existence of a phosphorus-rich layer, CTE mismatch is stronger than NiAu; therefore, the thermal fatigue resistance is weaker than NiAu. Due to the blocking effect of a thin nickel layer and the Ni–Sn–P layer, ultrathin ENEPIG can inhibit the growth of IMC and reduce the stress generation to a certain extent, but the failure risk under high cycles is higher than that of ENEPIG.

To observe the failed samples of ENEPIG 1 and OSP, the results are shown in Figure 20. The effect of thermal stress and strain on small-size solder balls was more serious, and the interface reaction was rapid, resulting in microcracks and voids. Therefore, the failure location was mostly concentrated on the side of the chip and substrate with a small bump size.

Due to the long-term temperature cycling process, a very obvious IMC layer was formed at the interface. Unlike the drop test, where cracks mostly occurred at the IMC interface, the failure in the high- and low-temperature cycle test occurred inside the solder near the IMC interface.

In the process of temperature cycling, the different CTE makes the volume change in each substance in the solder joint mismatched, which leads to stress and microcracks in the solder and promotes β- Sn grains recrystallize to form equiaxed grains [30], producing a continuous grain boundary network that provides a channel for the diffusion of microcracks. The crack extends with the volume expansion of microstructure, and finally leads to failure. It can be clearly seen from Figure 20 that the Sn grain boundary of the failure crack continued to extend, wherein the crack of OSP surface finish almost runs through the whole solder ball/IMC interface, while that of ENEPIG 1 diffuses upward along the grain boundary.

## 4. Conclusions

This study primarily focused on the interface reaction process and reliability between ultrathin ENEPIG surface finishes with two different Ni layer thicknesses and SAC305 solder. The results showed that, although the Ni(P) layer in ultrathin ENEPIG was eventually completely consumed, the Ni–Sn–P layer generated by an interface reaction still had a certain blocking effect on Cu diffusion. After aging for 1000 h, the thickness of brittle IMC was much smaller than that of OSP.

The shear test of solder ball showed that although the bonding force of ultrathin ENEPIG was weaker than that of standard ENEPIG, it was better than electroplated NiAu and OSP surface finish. Ultrathin ENEPIG produced a certain strength bonding with solder at the interface. In addition, the ultrathin ENEPIG with a thickness of 0.185 μm could pass the drop test. Finally, both types of ultrathin ENEPIG could reach the standard in TCT. Based on the above experimental results, it can be concluded that the ultrathin ENEPIG can meet the reliability standards of actual needs and has a broad application prospect in the field of high-frequency chip packaging.

## Figures and Tables

**Figure 1 materials-14-07874-f001:**
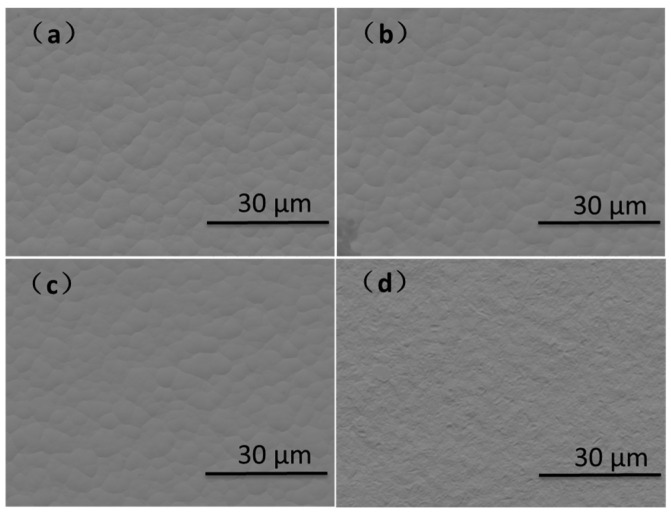
Surface morphology of different coatings. (**a**) ENEPIG 1; (**b**) ENEPIG 2; (**c**) ENEPIG 3; (**d**) electroplated Ni/Au.

**Figure 2 materials-14-07874-f002:**
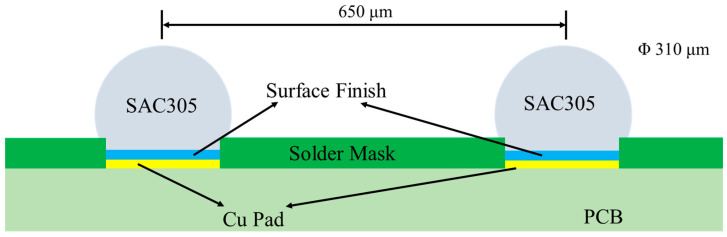
Structure of solder joints.

**Figure 3 materials-14-07874-f003:**
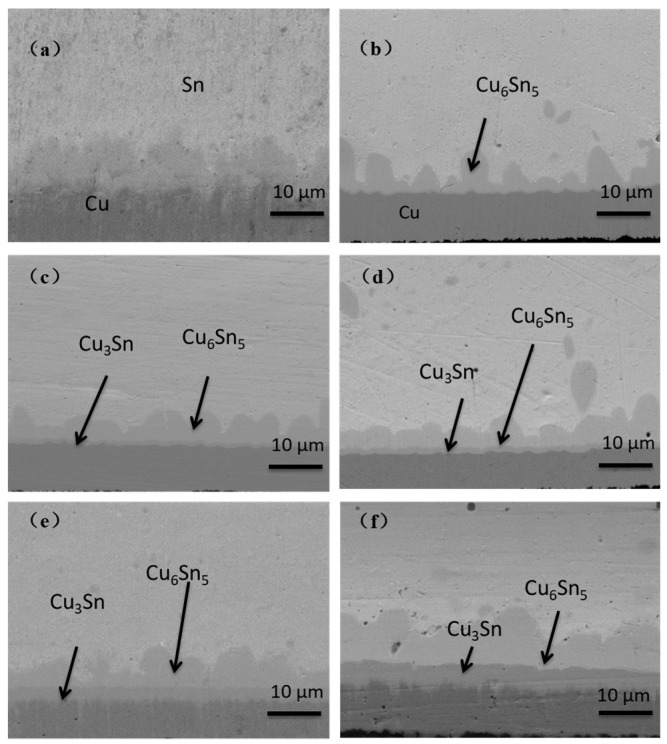
Evolution of IMC interface reaction on OSP surface finish with different aging times. (**a**) As reflowed; (**b**) 16 h; (**c**) 64 h; (**d**) 120 h; (**e**) 256 h; (**f**) 500 h.

**Figure 4 materials-14-07874-f004:**
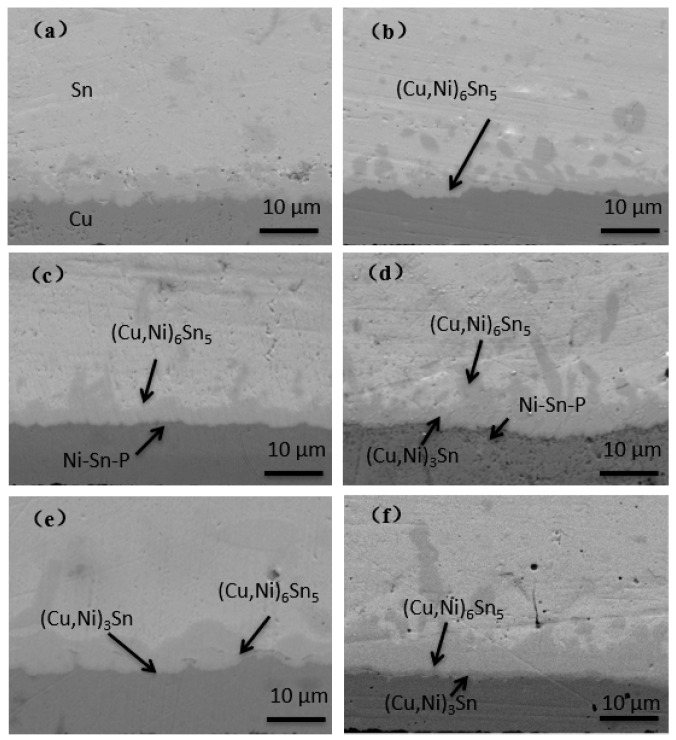
Evolution of IMC interface reaction on ENEPIG 1 surface finish with different aging times. (**a**) As reflowed; (**b**) 16h; (**c**) 64h; (**d**) 120 h; (**e**) 256 h; (**f**) 500 h.

**Figure 5 materials-14-07874-f005:**
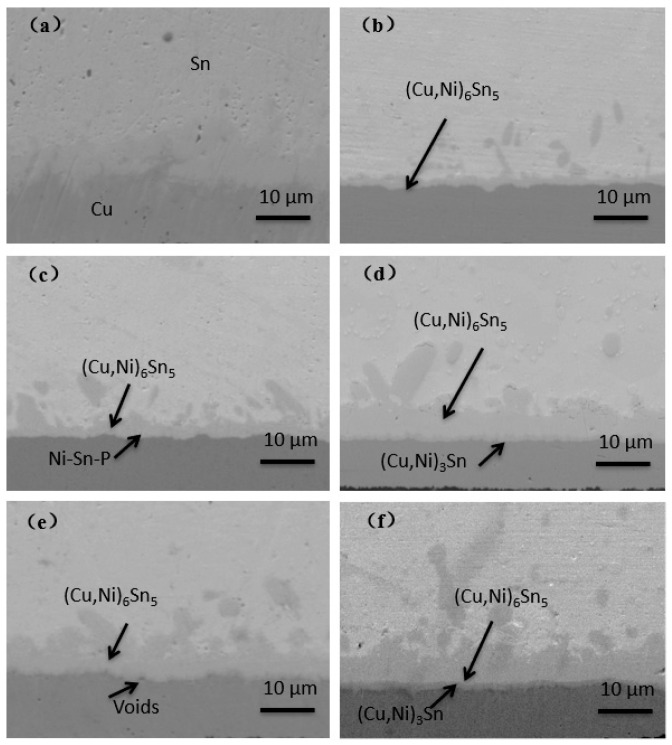
Evolution of IMC interface reaction on ENEPIG 2 surface finish with different aging times. (**a**) As reflowed; (**b**) 16 h; (**c**) 64 h; (**d**) 120 h; (**e**) 256 h; (**f**) 500 h.

**Figure 6 materials-14-07874-f006:**
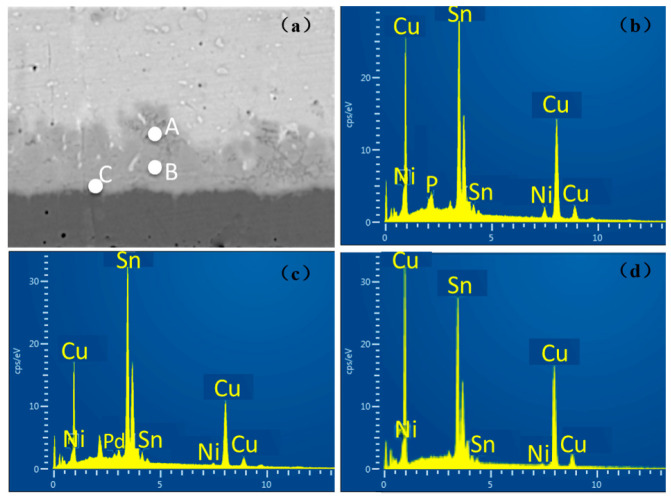
EDS results of IMC on ENEPIG 2 after 120 h aging. (**a**) Interface selection; (**b**) point A; (**c**) point B; (**d**) point C.

**Figure 7 materials-14-07874-f007:**
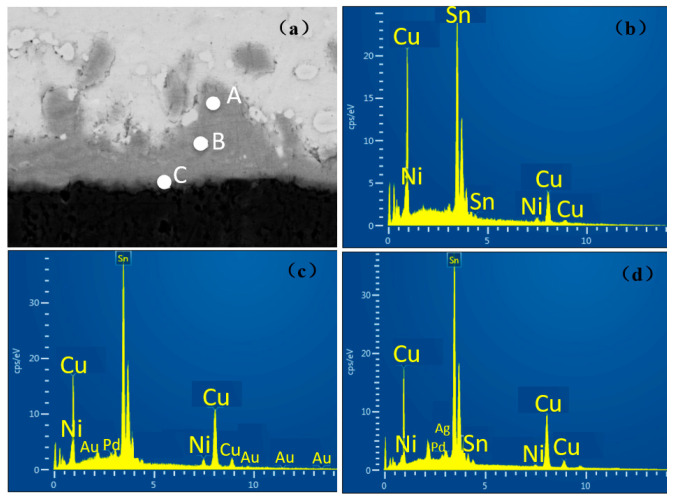
EDS results of IMC on ENEPIG 2 after 256 h aging. (**a**) Interface selection; (**b**) point A; (**c**) point B; (**d**) point C.

**Figure 8 materials-14-07874-f008:**
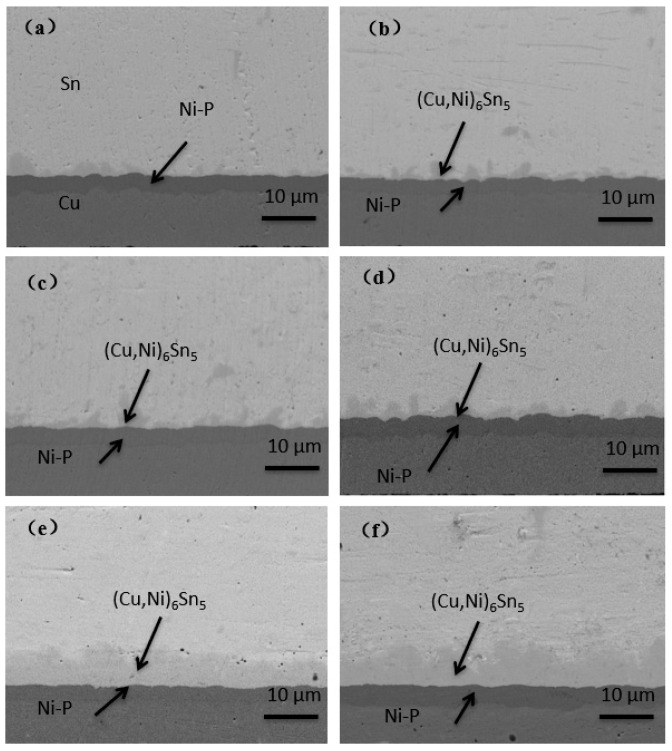
Evolution of IMC interface reaction on ENEPIG 3 surface finish with different aging times. (**a**) As reflowed; (**b**) 16 h; (**c**) 64 h; (**d**) 120 h; (**e**) 256 h; (**f**) 500 h.

**Figure 9 materials-14-07874-f009:**
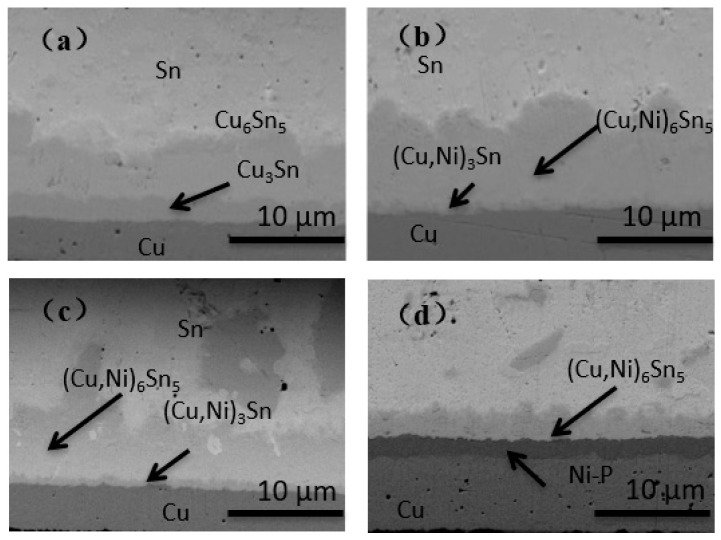
Cross-sections of different surface finishes after aging for 1000 h. (**a**) OSP; (**b**) ENEPIG 1; (**c**) ENEPIG 2; (**d**) ENEPIG 3.

**Figure 10 materials-14-07874-f010:**
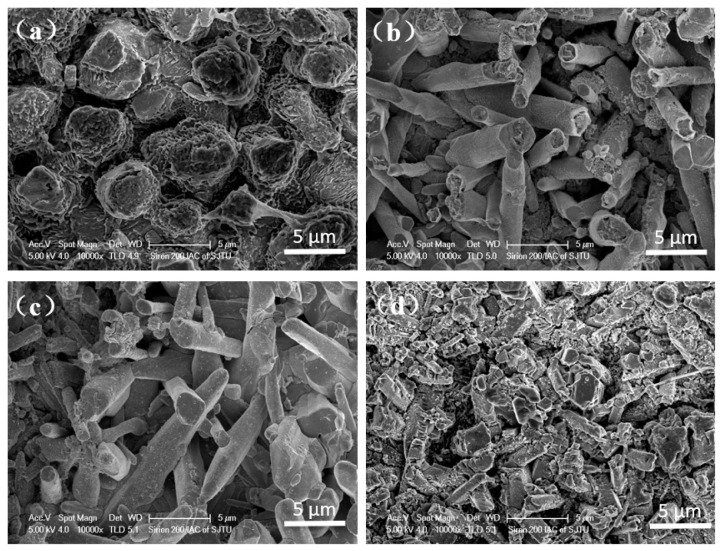
Surface morphology of IMC with different surface finishes after aging for 64 h. (**a**) OSP; (**b**) ENEPIG 1; (**c**) ENEPIG 2; (**d**) ENEPIG 3.

**Figure 11 materials-14-07874-f011:**
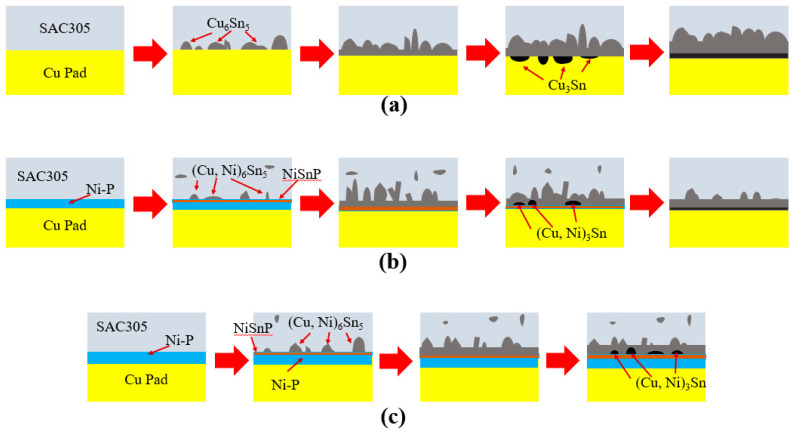
Evolution models of IMC under different surface finishes. (**a**) OSP; (**b**) ultrathin ENEPIG; (**c**) standard ENEPIG.

**Figure 12 materials-14-07874-f012:**
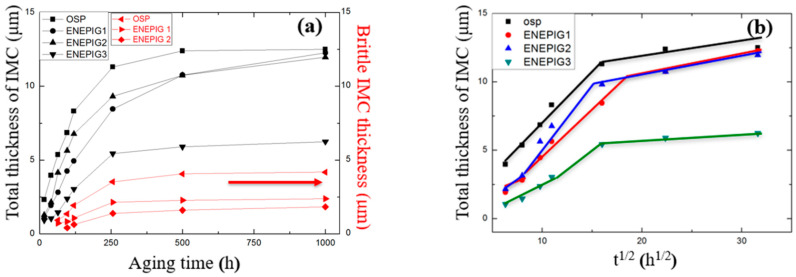
(**a**) Relationship curve between IMC growth and aging time in ENEPIG; (**b**) relationship curve between IMC growth and the square root of aging time in ENEPIG.

**Figure 13 materials-14-07874-f013:**
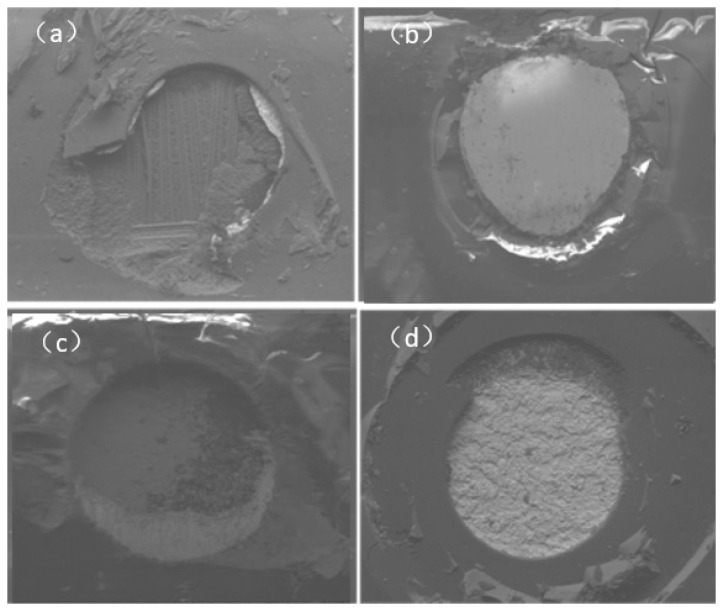
Schematic diagram of shear failure modes of weld balls. (**a**) Mode A: pad pull-out; (**b**) Mode B: ductile fracture in solder balls; (**c**) Mode C: fracture in ball/IMC interface; (**d**) Mode D: fracture in IMC interface.

**Figure 14 materials-14-07874-f014:**
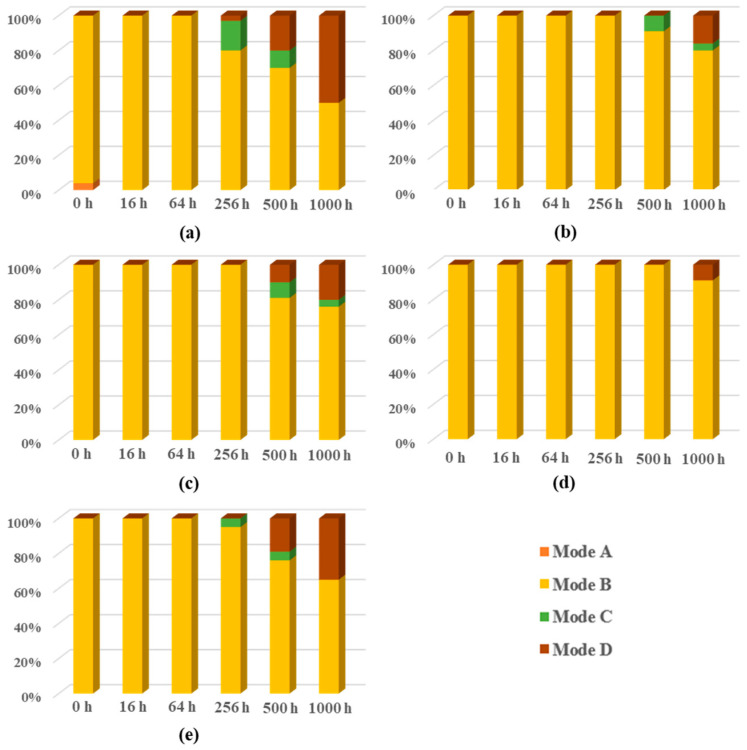
Failure mode statistics of shear experiment in different aging stages. (**a**) OSP; (**b**) ENEPIG 1; (**c**) ENEPIG 2; (**d**) ENEPIG 3; (**e**) Ni/Au.

**Figure 15 materials-14-07874-f015:**
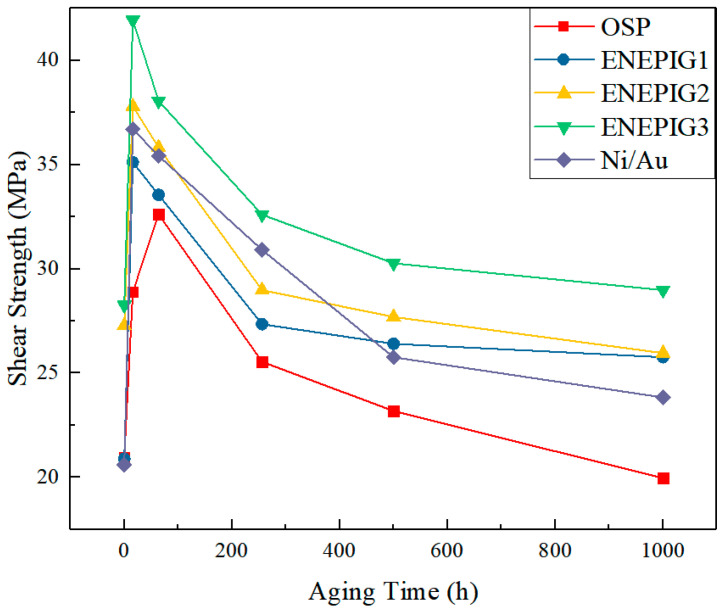
Relationship between shear strength and aging time for different surface finishes.

**Figure 16 materials-14-07874-f016:**
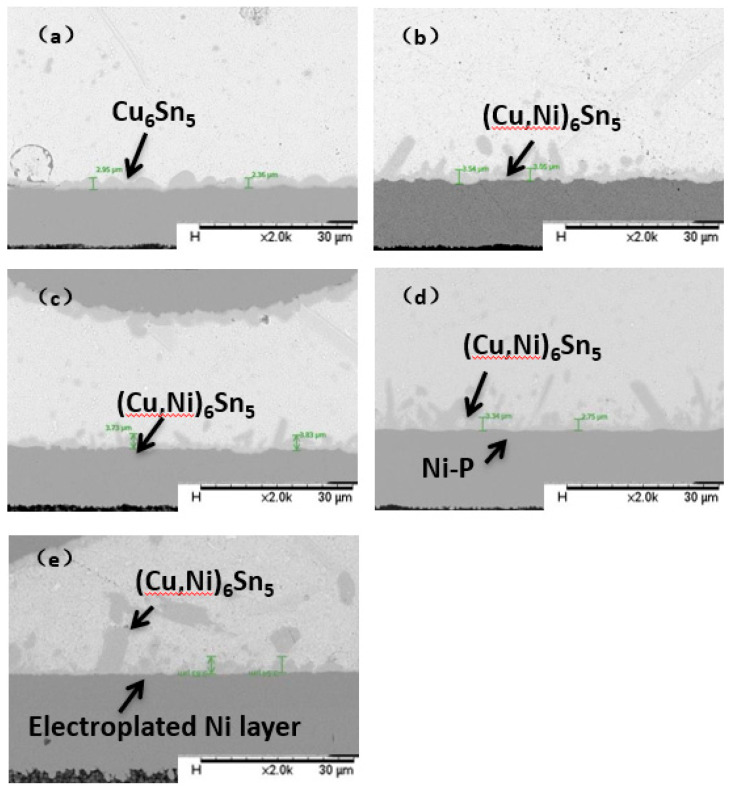
Cross-section of IMC growth of different surface finishes after drop test. (**a**) OSP; (**b**) ENEPIG 1; (**c**) ENEPIG 2; (**d**) ENEPIG 3; (**e**) Ni/Au.

**Figure 17 materials-14-07874-f017:**
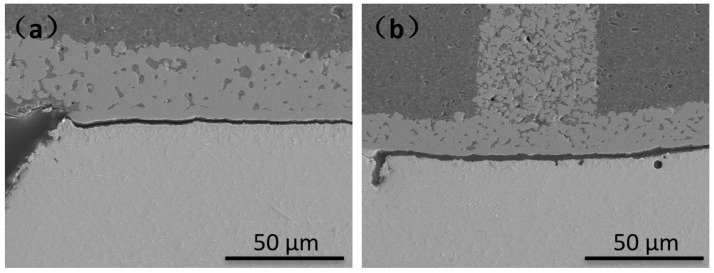
Cross-sectional view of the failed solder balls in drop test. (**a**) ENEPIG 1; (**b**) Ni/Au.

**Figure 18 materials-14-07874-f018:**
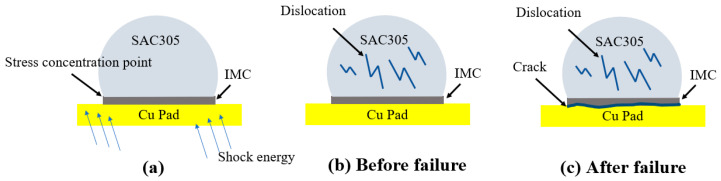
Failure mechanism of the drop test.

**Figure 19 materials-14-07874-f019:**
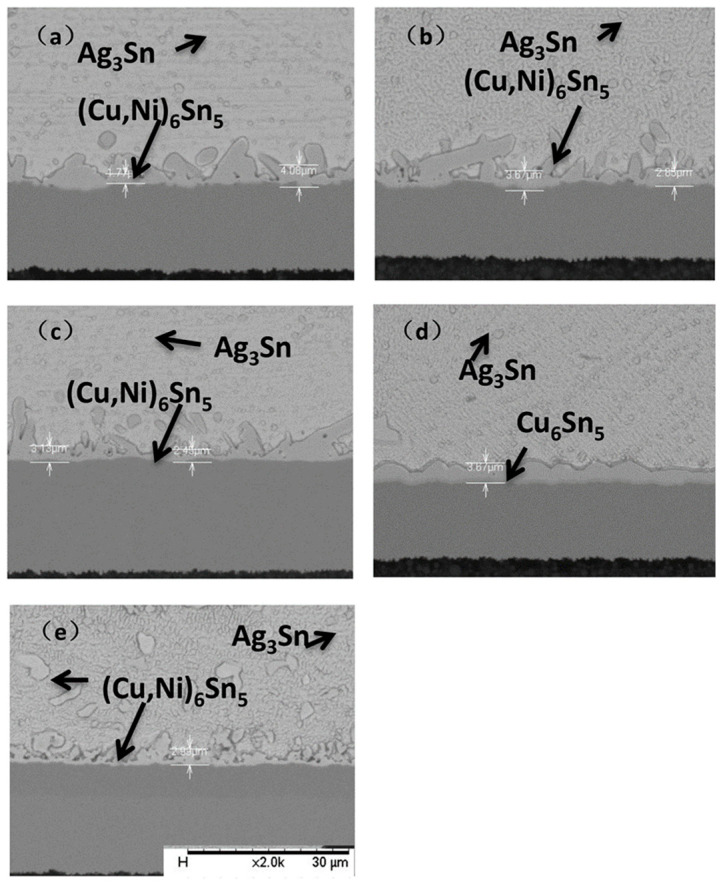
Cross-section of IMC growth of different surface finishes after TCT. (**a**) OSP; (**b**) ENEPIG 1; (**c**) ENEPIG 2; (**d**) ENEPIG 3; (**e**) Ni/Au.

**Figure 20 materials-14-07874-f020:**
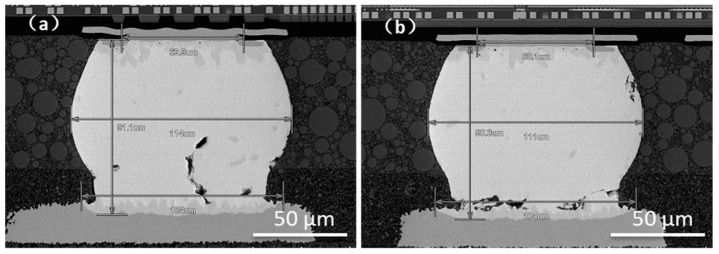
Cross-sectional view of the failed solder ball in TCT test. (**a**) ENEPIG 1; (**b**) OSP.

**Table 1 materials-14-07874-t001:** Coating thickness measurements for different surface finishes.

Surface Finish	Actual Value of Coating Thickness (μm)		
	Ni	Pd	Au
ENEPIG 1	0.112	0.093	0.088
ENEPIG 2	0.185	0.083	0.08
ENEPIG 3	4.61	0.115	0.052
Ni/Au	7.22	—	0.385

**Table 2 materials-14-07874-t002:** Composition of IMC on ENEPIG 2 after aging for 120 h.

Position	Elementary Composition (wt%)					IMC
	Cu	Ni	Sn	P	Pd	
A	45.57%	3.65%	50.62%	0.16%	-	(Cu, Ni)_6_Sn_5_
B	36.82%	0.85%	61.32%	-	1.01%	(Cu, Ni)_6_Sn_5_
C	66.8%	1.0%	32.2%	-	-	(Cu, Ni)_3_Sn

**Table 3 materials-14-07874-t003:** Different surface finishes’ drop test results.

Surface Finish	Pass Or Not	Cycles When Failed	Spalling or Not
OSP	Yes	-	No
ENEPIG 1	No	20	Yes
ENEPIG 2	Yes	-	No
ENEPIG 3	Yes	-	No
Ni/Au	No	15	Yes

**Table 4 materials-14-07874-t004:** Different surface finish TCT results.

Surface Finish	Pass or Not	Cycles when Failed
OSP	Yes	1124
ENEPIG 1	Yes	1602
ENEPIG 2	Yes	-
ENEPIG 3	Yes	-
Ni/Au	Yes	-

## Data Availability

The data presented in this study are openly available in the article.

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
