# Peer review of "Effect of Ni(P) Layer Thickness on Interface Reaction and Reliability of Ultrathin ENEPIG Surface Finish"

_materials, 2021, doi:10.3390/ma14247874_

Round 1
Reviewer 1 Report
The topic of the article is interesting, the presentation is appropriate. Two general comments and suggestions:
1. The quality of SEM images is good, but unfortunately the visibility of IMC layers is low in many cases. e.g. Fig7.c: (Cu, Ni)3Sn and Fig.7d: Ni-P, etc. These need to be improved, higher contrast needs to be applied.
2. Since the whole article examines the different mechanical properties as a function of the thickness of the IMC layers created by different finishing methods and thus classifies the methods, it is essential to quantify the IMC layer thickness accurately, calculated by measurement.
For example, in Fig. 14, the determination of the layer thickness is very poor, not accepted. A much more professional definition is required. With the help of image analysis software, the thickness of the IMC layer can be determined accurately from the ratio of the area and the length of the layer.
This exact definition of the IMC layer thickness is an essential condition for the publication of the article, so it is recommended to carry out this. The additional modification will greatly enhance the quality of the article.
Reviewer 2 Report
The authors studied the interface reaction process and reliability between ultrathin ENEPIG surface finishes with two different Ni layer thicknesses and SAC305 solder. The results showed that although the Ni(P) layer in ultrathin ENEPIG was finally completely consumed, the Ni-Sn-P layer generated by interface reaction still had a certain blocking effect on Cu diffusion.
The authors provided abundant data for the conclusion, but there’s a concern for the composition identification. Fig.3-Fig.7, the author mentioned the intermetallic materials but didn’t explain how they knew the composition of these materials. The authors should give a clear description of the material characterizations. The same issue should be treated for Fig. 14 and Fig. 17.
The paper can be published after major revision.
Round 2
Reviewer 1 Report
Thanks for the corrections, which improve the quality of the article!
Author Response
We appreciate your efforts to help us improve the quality of our manuscript. Your comments are valuable and very helpful for revising and improving our paper!